# Proteomic and Phosphoproteomic Analysis Reveals Differential Immune Response to Hirame Novirhabdovirus (HIRRV) Infection in the Flounder (*Paralichthys olivaceus*) under Different Temperature

**DOI:** 10.3390/biology12081145

**Published:** 2023-08-18

**Authors:** Xiaoqian Tang, Yingfeng Zhang, Jing Xing, Xiuzhen Sheng, Heng Chi, Wenbin Zhan

**Affiliations:** 1Laboratory of Pathology and Immunology of Aquatic Animals, KLMME, Ocean University of China, Qingdao 266003, China; tangxq@ouc.edu.cn (X.T.); zhang_yf0921@163.com (Y.Z.); xingjing@ouc.edu.cn (J.X.); xzsheng@ouc.edu.cn (X.S.); chiheng@ouc.edu.cn (H.C.); 2Laboratory for Marine Fisheries Science and Food Production Processes, Qingdao National Laboratory for Marine Science and Technology, Qingdao 266071, China

**Keywords:** Hirame novirhabdovirus (HIRRV), temperature, proteomics, phosphoproteomics, immune responses

## Abstract

**Simple Summary:**

The outbreak of Hirame novirhabdovirus is significantly temperature dependent. The aim of this study was to identify differential responses of flounder to HIRRV infection at different temperatures by proteome and phosphoproteome. Post HIRRV infection under 10 °C and 20 °C, the enriched immune-related DEPs were both involved in RLR and NLR signaling pathways, apoptosis, phagosome and lysosome, and the DEPPs were also both enriched in spliceosome, mTOR signaling pathway and RNA transport. Compared with the group under 10 °C, the proteins and phosphoproteins involved in interferon production and signaling showed stronger response to infection under 20 °C. qRT-PCR assay showed that eight antiviral-related mRNA including IRF3, IRF7, IKKβ, TBK1, IFIT1, IFI44, MX1 and ISG15 displayed significantly stronger and quicker response at early infection under 20 °C. This study provided a comprehensive understanding of signaling alterations and differential antiviral responses of flounder to HIRRV infection under different temperatures.

**Abstract:**

Hirame novirhabdovirus (HIRRV) is one of most serious viral pathogens causing significant economic losses to the flounder (*Paralichthys olivaceus*)-farming industry. Previous studies have shown that the outbreak of HIRRV is highly temperature-dependent, and revealed the viral replication was significantly affected by the antiviral response of flounders under different temperatures. In the present study, the proteome and phosphoproteome was used to analyze the different antiviral responses in the HIRRV-infected flounder under 10 °C and 20 °C. Post viral infection, 472 differentially expressed proteins (DEPs) were identified in the spleen of flounder under 10 °C, which related to NOD-like receptor signaling pathway, RIG-I-like receptor signaling pathway, RNA transport and so on. Under 20 °C, 652 DEPs were identified and involved in focal adhesion, regulation of actin cytoskeleton, phagosome, NOD-like receptor signaling pathway and RIG-I-like receptor signaling pathway. Phosphoproteome analysis showed that 675 differentially expressed phosphoproteins (DEPPs) were identified in the viral infected spleen under 10 °C and significantly enriched in Spliceosome, signaling pathway, necroptosis and RNA transport. Under 20 °C, 1304 DEPPs were identified and significantly enriched to Proteasome, VEGF signaling pathway, apoptosis, Spliceosome, mTOR signaling pathway, mRNA surveillance pathway, and RNA transport. To be noted, the proteins and phosphoproteins involved in interferon production and signaling showed significant upregulations in the viral infected flounder under 20 °C compared with that under 10 °C. Furthermore, the temporal expression profiles of eight selected antiviral-related mRNA including IRF3, IRF7, IKKβ, TBK1, IFIT1, IFI44, MX1 and ISG15 were detected by qRT-PCR, which showed a significantly stronger response at early infection under 20 °C. These results provided fundamental resources for subsequent in-depth research on the HIRRV infection mechanism and the antiviral immunity of flounder, and also gives evidences for the high mortality of HIRRV-infected flounder under low temperature.

## 1. Introduction

Japanese flounder (*Paralichthys olivaceus*) is one of the most economically important aquaculture fishes. In recent years, viral diseases outbreaks have significantly affected farmed flounder, especially *Hirame novirhabdovirus* (HIRRV), which caused serious economic losses [1,2]. HIRRV, as a negative-stranded RNA virus belonging to Novirhabdovirus genus within the Rhabdoviridae family, has been reported to infect many marine and freshwater fish species, such as black seabream (*Acanthopagrus schlegeli*) [3], sea bass (*Lateolabrax maculatus*) [4], stone flounder (*Kareius bicoloratus*) [5], grayling (*Thymallus thymallus*) [6] and brown trout (*Salmo trutta*) [7]. Although the severe losses of flounder due to HIRRV infections have prompted the development of vaccines to curb the disease [8,9,10], no effective commercial vaccine is available for worldwide use till now, and there are also no effective drugs available against HIRRV. Hence, a deep understanding of the HIRRV infection mechanism and antiviral immune response of flounder would provide valuable clues and useful guidance for the prevention and treatment of HIRRV.

It was well documented that water temperature can affect the viral pathogenicity and disease progression in fish. HIRRV is mainly prevalent below 15 °C, with a peak incidence around 10 °C, and this disease has a tendency to heal on its own when the water temperature is above 15 °C [11]. Similarly, disease occurrence caused by several fish rhabdoviruses, including infectious hematopoietic necrosis virus (IHNV) [12], viral hemorrhagic septicaemia virus (VHSV) [13] and spring viraemia of carp virus (SVCV) [14], were also found to be significantly temperature-dependent. Our previous study further revealed that HIRRV could effectively propagate in the epithelioma papulosum cyprini (EPC) cells both at 10 °C and 20 °C, whereas the artificial infection rarely induced the mortality of HIRRV-infected flounder under 20 °C [15]. Therefore, it is plausible to assume that a significantly stronger antiviral immune response was triggered in the flounder cultured under 20 °C, which effectively inhibit virus proliferation. It is well known that water temperature could significantly influence the immunity of fish [16], and exposure to sub-optimal temperature would generally have suppressive effects on the innate and adaptive immune systems of fishes [17]. It was reported that VHSV mediated down-regulation of TLR2 and TLR7 in the flounder at low temperature, causing an extended inflammatory condition and delayed response of interferon-stimulated genes (ISGs) [18]. In the Atlantic cod injected with viral mimics, the expression of genes involved in NF-κB and type I interferon signaling pathways were significantly regulated by temperature [19]. Our previous study demonstrated that HIRRV was able to infect mIgM^+^ B lymphocytes of flounder, but the number of viral copies in 10 °C infection group were significantly higher than that in 20 °C infection group [20]. Our recent study further found that RLR signaling was significantly activated in the HIRRV-infected flounder under 20 °C compared with that at 10 °C [15]. Although significant research progress was achieved on HIRRV over the past years, the mechanisms of the viral pathogenesis and antiviral immunity in host fish are yet to be deciphered.

Proteins are the performers of life activities and the abundance of mRNA may not accurately predict the amount of corresponding functional proteins, while proteomic approaches can provide accurate clues about biological complexity [21,22]. Phosphoproteomic analysis could provide further information on the signal transduction and regulation of gene expression in antiviral immune response [23,24]. Thus, integrated proteomic and phosphoproteomic analysis would give a deep insight into the immune defense mechanism of host against viral infection and facilitate our understanding of viral pathogenesis [25]. Reversible protein phosphorylation plays a role in signal transduction and subcellular localization of proteins, in addition to its influence on protein structure [26,27]. Many studies of host proteins after viral infection confirm significant changes in their phosphorylation levels [28,29]. However, the application of phosphoproteomic analysis in fish is relatively lacking. Dynamic changes in protein phosphorylation patterns triggered by bacterial infection were obtained after analysis of phosphatomic expression profiles of Atlantic salmon kidneys challenged with *A. salmonicida* [30]. The unbiased and system-wide analysis of the quantitative proteome and phosphoproteome in gibel carp investigated the global phosphorylation and protein abundance landscape of Cyprinid Herpesvirus 2 (CyHV-2)-infected gibel carp, which provided us with a comprehensive understanding of CyHV-2 infection [31]. The comparative phosphoproteomics and transcriptomics of black carp (*Mylopharyngodon piceus*) kidney cells with/without grass carp reovirus (GCRV) infection explored the crucial genes and pathways involved in the antiviral innate immunity of black carp and contributed as an indispensable reference map for the further investigation of the innate immune system of black carp [32]. However, at present, there are no comprehensive proteomics and phosphoryation proteomics analysis of the flounder infected with HIRRV.

In the present study, iTRAQ LC-MS/MS were used to determine the differences in protein and phosphorylated proteins levels in flounder challenged by HIRRV under different water temperatures. In proteomics analysis, a total of 4952 proteins were identified and quantified, and we quantified the phosphorylation status of around 10,442 different phosphorylation sites from 3796 proteins based on the phosphoproteomics analysis. Then, we analyzed the relationship between differential proteins and HIRRV infection to find their role in the process of HIRRV infection. These results would lay a foundation for understanding of the different antiviral mechanism against HIRRV in flounder under different temperatures.

## 2. Materials and Methods

### 2.1. Experimental Fish and Virus

Healthy flounder (22 ± 3 cm) were purchased from a fish farm, located in Rizhao, Shandong Province. The fish were raised in automatic water cycler system (500 L) supplied with aerated and filtrated water under 16 °C. Farming water conditions were monitored by the YSI multi-probe system (YSI 556) to meet the following conditions: dissolved oxygen 6.0 ± 0.5 mg/L, pH = 7.0, total ammonia <0.2 mg/L and nitrite <0.02 mg/L. Fish were selected to ensure that they were free of HIRRV and other viral infections. All experimental fish were fed with commercial dry pellets for two weeks prior to commencing experimental work.

The HIRRV CNPo2015 strain used in this study was kept by our laboratory at −80 °C [33]. Epithelioma papulosum cyprinid (EPC) cells were used for viral propagation, cultured in M199 medium with 2% fetal bovine serum and 1% penicillin-streptomycin (Gibco, Waltham, MA, USA) at 20 °C. The cell suspension was collected by repeated freezing and thawing three times to allow the virus to exit the cell, and centrifugation to remove cellular debris, viral titer was adjusted to 1.0 × 10^7.8^ TCID_50_/mL prior to flounder injection.

### 2.2. Experimental Infection and Sampling

Following acclimation, the water temperatures were gradually adjusted into 10 °C and 20 °C within 72 h. The flounder under different temperatures were subdivided into infection group injected with 100 μL HIRRV stock (1.0 × 10^7.8^ TCID_50_/mL) and control group injected with 100 μL culture supernatant (without HIRRV) separately. So, in this study, we set up four groups with 70 fish in each. For proteome and phosphoproteome, after 24 h of infection, five flounders were randomly selected from each group, and the spleen of flounder was sampled and stored in liquid nitrogen for later use. To reduce fish pain and stress, the experimental fish were anesthetized with MS-222 before injection and sampling. All experimental methods were approved by the Instructional Animal Care and Use Committee of the Ocean University of China (permit number: 20150101). For the sake of simplicity, the viral infection groups under 10 °C and 20 °C are abbreviated as S10-I and S20-I, respectively, and S10-C and S20-C represent two control groups. For RT-qPCR, spleen was collected from five fish in each group at 7 different time points (0, 6, 12, 24, 48, 72 and 96 h) after injection, RNAlater (Takara, Dalian, China) was chosen to dissolve the samples, then stored at −80 °C for RNA extraction.

### 2.3. Protein Extraction and iTRAQ Labeling

The fish spleen was grinded by liquid nitrogen into cell powder and dissolve the powder into the lysis buffer (8 M urea, 2 mM EDTA, 10 mM DTT and 1% Protease Inhibitor Cocktail III). For PTM experiments, inhibitors were added to the lysis buffer with the phosphatase inhibitor PhosSTOP (Roche, Basel, Switzerland).

The above solutions were broken by sonication three times on ice using a high intensity ultrasonic processor (Scientz, Ningbo, China). After centrifugation of the fragmented sample at 12,000× *g* for 10 min at low temperature, the precipitate was removed and the protein concentration was determined using a BCA kit. The protein solution was then enzymatically digested into peptides in the following steps. First, the protein solution was reduced by adding an appropriate amount of dithiothreitol (DTT, Sigma, St. Louis, MO, USA) to a final concentration of 5 mM for 30 min at 56 °C. After that, iodoacetamide (IAA, Sigma, USA) was added to a final concentration of 11 mM to alkylated and incubated for 15 min at room temperature and protected from light. To ensure the quality of the samples, the concentration of urea in the samples was diluted with 100 mM TEAB (Sigma, USA) to less than 2 M. Digestion was performed by adding trypsin to the protein solution in batches. First, trypsin was added to the proteins at a mass ratio of 1:50 (trypsin/protein) and digested for 12 h at 37 °C. Subsequently, trypsin was further added at a mass ratio of 1:100, and the digestion was continued for 4 h.

Peptides were desalted using Strata X C18 SPE column (Phenomenex, Torrance, CA, USA) and vacuum-dried. The peptides were subsequently labeled with iTRAQ. Briefly, one unit of iTRAQ reagent was thawed in acetonitrile. The peptide mixtures were then incubated for 2 h with iTRAQ reagent at room temperature and pooled, desalted and dried by vacuum centrifugation. The eluted peptides were collected and dried by vacuum centrifugation of the peptide mixture.

### 2.4. HPLC Fractionation

The tryptic peptides were fractionated into fractions by high pH reverse-phase HPLC using Thermo Betasil C18 column (Thermo Electron Corporation, Waltham, MA, USA). Briefly, peptides were first separated with a gradient of 8% to 32% acetonitrile (pH 9.0) over 60 min into 60 fractions. The peptides were combined into 8 fractions and dried by vacuum centrifuging, the peptides were then dissolved in enrichment buffer solution (50% acetonitrile/6% trifluoroacetic acid) and the supernatant was transferred to the immobilized metal-chelating affinity chromatography (IMAC) material, which was washed in advance and incubated with gentle shaking on a rotary shaker. After incubation, the resin was washed three times with buffer solution (containing 50% acetonitrile, 6% trifluoroacetic acid and 30% acetonitrile/0.1% trifluoroacetic acid), and finally, the modified peptide was eluted with 10% ammonia, the eluate was collected and vacuum frozen and dried and the C18 Zip Tips kit (Millipore Corp, Billerica, MA, USA) was used for desalting.

### 2.5. Affinity Enrichment

To enrich modified peptides, peptide mixtures were first incubated with IMAC microspheres suspension with vibration in loading buffer (50% acetonitrile/6% trifluoroacetic acid). The IMAC microspheres with enriched phosphopeptides were collected by centrifugation, and the supernatant was removed. To remove nonspecifically adsorbed peptides, the IMAC microspheres were washed with 50% acetonitrile/6% trifluoroacetic acid and 30% acetonitrile/0.1% trifluoroacetic acid, sequentially. The IMAC microspheres were then added to elution buffer containing 10% NH_4_OH and shaken to elute the enriched phosphopeptides. The supernatant containing the phosphopeptides was collected and lyophilized for LC-MS/MS analysis.

### 2.6. LC-MS/MS Analysis and Data Analysis

Peptides were eluted using concentration gradient variation (*v*/*v*) of acetonitrile in 0.1% (*v*/*v*) formic acid (FA), and then dried to completion for tandem mass spectrometric (LC-MS/MS) analysis. The peptides were subjected to NSI source followed by tandem mass spectrometry (LC-MS/MS) in Q ExactiveTM Plus (Thermo, Carlsbad, CA, USA) coupled online to the UPLC.

The resulting LC-MS/MS data were processed using Maxquant search engine (v.1.5.2.8). False discovery rate (FDR) was adjusted to <1% and minimum score for modified peptides was set to 40. The screening criteria for splenic differentially expressed proteins (DEPs) between the epithelioma papulosum cyprinid (EPC) supernatant and HIRRV-infected groups in 10 °C and 20 °C were *p* ≤ 0.05 and expression fold change >1.2. The screening criteria for splenic differentially expressed phosphoproteins (DEPPs) between the EPC supernatant and HIRRV-infected groups in 10 °C and 20 °C were *p* ≤ 0.05 and expression fold change >1.5. The results of the obtained mass spectrometry data were analyzed using MaxQuant software (version 1.5). The protein expression level in infected groups was normalized to the control groups under the same temperature. GO analysis describes the function of proteins in three categories: biological process, cellular composition and molecular function. Cellular composition refers to the specific components of a cell. Molecular function mainly describes the chemical activity of a molecule, such as catalytic activity or binding activity that can be expressed at the molecular level. An orderly series of molecules in an organism performing a specific function is known as biological process. First, the protein ID was converted to UniProt ID by the UniProt-GOA database (http://www.ebi.ac.uk/GOA/ (URL (accessed on 15 March 2021) system, after which the UniProt ID was used to match the GO ID and analyzed based on the GO ID. If the UniProt-GOA database did not contain information about the query protein, the InterProScan software was used to predict the GO function of the protein. Kyoto Encyclopedia of Genes and Genomes (KEGG) database was used to annotate protein pathway. KEGG pathways mainly included: metabolism, genetic information processing, environmental information processing, cellular processes, rat diseases and drug development. For the pathways involved, we annotated them with the KEGG pathway database: first, the KEGG online service tool KAAS was used to annotate with the submitted proteins, and KEGG mapper was then used to match the annotated proteins to the corresponding pathways in the database.

### 2.7. Quantitative Real-Time Polymerase Chain Reaction

Based on the identification result of proteomic and phosphoproteomic analysis, eight antiviral-related mRNA, including ISG15, IRF3, IRF7, IKKβ, TBK1, IFIT1, IFI44 and MX, were further screened, and their temporal expression profiles were examined by reverse transcription quantitative real-time PCR (RT-qPCR). The specific primers for above eight genes were designed using Primer Premier software (Version 5.0) and listed in Table 1, we chose 18s rRNA as the internal reference gene. The qRT-PCR was performed on LightCycler^®^ 480 II Real Time System (Roche) using SYBR GreenIMaster (ABM), and the expression levels of selected genes were analyzed by the 2^−ΔΔCt^ method [34].

### 2.8. Statistical Analysis

The data obtained were analyzed using Statistical Product and Service Solution (SPSS) software (Version 20.0, IBM). The statistical significance of the immune-related proteins transcriptional level in flounder under different temperatures was analyzed by unpaired *t*-test. Values were considered as significant at *p* < 0.05.

## 3. Results

### 3.1. Analysis of DEPs and DEPPs under 10 °C

When comparing the HIRRV-infected group with the control group under 10 °C, there were 472 DEPs identified with 278 down- and 194 up-regulated proteins (Figure 1A). Compared with the control group, 59 DEPPs were up-regulated and 616 DEPPs were down-regulated (Figure 1B).

The GO classification showed that the DEPs between S10-I and S10-C groups were mainly involved in the cellular process, metabolic process and single-organism process in the category of Biological Process. In the category of Cellular Component, the mainly enriched subcategories were Cell, Organelle and Macromolecular complex. In Molecular Function, most DEPs were enriched in the Binding and Catalytic activity subcategories (Figure 2A). The KEGG pathway analysis showed that the DEPs between S10-I and S10-C were significantly enriched in carbon metabolism, salmonella infection, ribosome, ECM–receptor interaction and tight junction within different categories. To be noted, the DEPs were also enriched in some immune-related pathways, such as NOD-like receptor signaling pathway, Herpes simplex infection, spliceosome, calcium signaling pathway, phagosome and apoptosis. Most proteins in these pathways were down-regulated, such as the proteins involved in the RLR and NLR pathways, suggesting that antiviral pathways were inhibited during HIRRV infection at 10 °C (Figure 2B). KEGG enrichment analysis was also performed for the DEPPs between S10-I and S10-C, and the result revealed that the DEPPs were enriched into four pathways, including spliceosome, necroptosis, mTOR signaling pathway and RNA transport. Interestingly, the phosphorylation level of the proteins involved in these enriched signaling pathways were all reduced at 10 °C (Figure 2C). Additionally, a domain enrichment analysis was carried out, and 16 protein domains were identified in the DEPPs between S10-I and S10-C, such as RNA recognition motif domain, nucleotide-binding alpha–beta plait domain, SH2 domain, SPOC domain and RAS-associating domain. (Figure 2D).

### 3.2. Analysis of DEPs and DEPPs under 20 °C

Compared with the control group injected with EPC supernatant, there were 652 DEPs identified with 316 down- and 336 up-regulated proteins in the HIRRV-infected group under 20 °C (Figure 1A), which also showed that 1272 DEPPs were up-regulated and 32 DEPPs were down-regulated (Figure 1B).

The GO classification showed that the DEPs between S20-I and S20-C groups were mainly involved in the cellular process, metabolic process and single-organism process in the category of Biological Process. In the category of Cellular Component, the mainly enriched subcategories were Cell, Organelle and Macromolecular complex. In Molecular Function (MF), most DEPs were enriched in the Binding and Catalytic activity subcategories (Figure 3A). The KEGG pathway analysis showed that the DEPs between S20-I and S20-C were distributed in focal adhesion, regulation of actin cytoskeleton, ECM–receptor interaction, ribosome and protein processing in endoplasmic reticulum within different categories. To be noted, the DEPs were also enriched in some immune related pathways, such as RIG-I-like receptor signaling pathway, NOD-like receptor signaling pathway, salmonella infection, herpes simplex infection, spliceosome, calcium signaling pathway and phagosome. In contrast to the situation under 10 °C, most proteins in these enriched pathways were significantly up-regulated under 20 °C. For instance, the majority of proteins involved in RLR and NLR signaling pathways were up-regulated, suggesting that antiviral pathways were significantly activated post HIRRV infection at 20 °C (Figure 3B). KEGG enrichment analysis was also performed for the DEPPs between S20-I and S20-C, and the result revealed that the DEPPs were enriched into seven pathways, including proteasome, VEGF signaling pathway, apoptosis, spliceosome, mTOR signaling pathway, mRNA surveillance pathway, RNA transport. Notably, the phosphorylation level of proteins involved in these signaling pathways were all increased at 20 °C, which is also consistent with proteomics findings (Figure 3C). Additionally, a domain enrichment analysis was carried out, and 20 protein domains were identified in the DEPPs between S20-I and S20-C, such as SH2 domain, Small GTP-binding protein domain, Zinc finger, LIM-type, Armadillo-like helical, UBA-like and so on (Figure 3D).

### 3.3. Temperature Affects the Anti-HIRRV Response of Flounder

A Venn diagram was used to screen the specific DEPs and DEPPs that showed differential expression response to the HIRRV infection between under 10 °C and 20 °C. Excluding the effect of temperature on the flounder without infection, 192 DEPs and 318 DEPPs were found to be specifically up-regulated post HIRRV infection under 20 °C compared to that under 10 °C (Figure 4A and Figure 5A). The GO classification showed that the DEPs were mainly involved in the cellular process, metabolic process and single-organism process of Biological Process, Binding, Catalytic activity and Catalytic activity of Molecular Function. In the category of Cellular Component, most DEPs were mainly enriched in Cell, Cell part, Mocromolecular complex and Organelle (Figure 4B). These differentially expressed proteins were mainly involved in 23 immune-related pathways (Figure 4C), of which phagosome was the mostly enriched pathway, the leukocyte transendothelial migration, hepatitis C, complement and coagulation cascades, NOD-like receptor signaling pathway, antigen processing and presentation and other KEGG signaling pathways were also significantly enriched. We further screened 27 differentially expressed proteins associated with antigen recognition, processing, interferon production and inflammatory response. The expression levels of each protein in different treatments were shown by heat map analysis (Figure 4D). The screened DEPPs by Venn diagram were also analyzed in the same way. The GO classification analysis of these up-regulated DEPPs showed that they were mainly related to cellular processes, metabolic processes, single-organism processes of Biological Process. In terms of Molecular Function, Binding molecules, Catalytic activity and Structural molecular activity were highly enriched. With regard to Cellular Component, most DEPPs were associated with Cell, Cellpart and Organelle (Figure 5B). Most of the DEPPs were related to endocytosis, followed by the MAPK signaling pathway, NLR signaling pathway and chemokine signaling pathway (Figure 5C). Then, 19 DEPPs associated with antigen recognition, processing, interferon production and inflammatory response were further selected, and their expression profiles among four treatments are represented by heat map (Figure 5D).

### 3.4. Temporal Expressions of Genes Involved in Interferon Antiviral Response

In order to further analyze the antiviral response of flounder after HIRRV infection, expression profiles of eight immune-related genes involved in interferon production and action were detected by qRT-PCR, including two interferon regulatory factors (IRF3, IRF7), two key kinases (IKKβ, TBK1), three interferon-stimulating genes (IFIT1, IFI44, MX1, ISG15). Post HIRRV infection, all the eight detected mRNA were significantly induced both under 10 °C and 20 °C. It was further notably found that nearly all the detected mRNA showed significantly higher and quicker responses to HIRRV under 20 °C than that under 10 °C within 24 h post infection (hpi). In contrast, since 48 hpi, all the detected genes except TBK1 showed significantly higher expressions in the flounder under 10 °C compared with that under 20 °C (Figure 6).

## 4. Discussion

It is well documented that the environmental temperature influences adaptive and innate immunity of poikilothermic vertebrates, including adaptive and innate immunity [16]. Our previous transcriptome analysis found that after infection with HIRRV at 10 °C and 20 °C, the differentially expressed genes in flounder spleen were significantly enriched in inflammatory and immune-related pathways like cytokine–cytokine-receptor interaction, TLR signaling pathway, RLR signaling pathway, NLR signaling pathway and Cytosolic DNA-sensing pathway, etc. HIRRV infection at 20 °C can significantly stimulate and activate the RLRs pathway of flounder compared with that under 10 °C [15]. It was also found that HIRRV infection at 20 °C could significantly induce the up-regulation of the antigen processing and presentation pathway [20]. Similarly, in the present study, the proteome and phosphoproteome analysis showed that the antiviral immune response in flounder differed significantly at different temperatures post HIRRV infection. After HIRRV infection under 10 °C and 20 °C, the enriched immune-related DEPs were both associated in RLR and NLR signaling pathways, apoptosis, phagosome and lysosome, and the DEPPs were also both involved in spliceosome, mTOR signaling pathway and RNA transport. Notably, the proteins and phosphoproteins involved in interferon production and signaling showed a stronger response to HIRRV infection under 20 °C compared with that under 10 °C. qRT-PCR assay showed that eight antiviral-related mRNA displayed significantly stronger and quicker responses at early infection under 20 °C. These results demonstrated that the antiviral response of flounder to HIRRV infection under different temperature conditions are significantly different at gene, protein and phosphorylation levels.

Non-specific immunity is the first line of defense against pathogenic microorganisms, and after effective recognition of pathogen-associated molecule patterns (PAMPs), cellular pattern recognition receptors (PRRs) subsequently initiate downstream signaling to resist pathogen infections [35,36]. The TLRs, RLRs and NLRs are the main PRRs responsible for sensing viral infestation [37,38,39]. The binding of viral PAMPs to these PRRs triggers a signaling cascade that induces the expression of viral response genes and pro-inflammatory cytokines such as type I interferon (IFNs), thereby limiting viral replication and regulating adaptive immunity [40]. Transcriptome analysis showed that NLR signaling pathway of flounder was significantly enriched after megalocytivirus infection [41]. NLRs are a specialized group of intracellular receptors and function as surveillance sensors against microbial products and danger signals and, thereby, trigger host defense pathways [42]. Notably, one NLR family member, NLRC3, was found to be significantly downregulated at 10 °C but upregulated at 20 °C post HIRRV. In teleost, the NLRC3 of grouper acts as a host factor that negatively regulated the antiviral immune response to facilitate nervous necrosis virus replication [43]; however, zebrafish NLRC3-like protein could interact with SVCV, and has positive regulatory function during SVCV infection [44]. So, the regulatory role of flounder NLRC3 in antiviral immunity remains to be further elucidated.

RLR proteins are expressed in almost all cells, mainly in the cytoplasm, and specifically recognize virus-derived RNA species, thus distinguishing pathogens from their hosts [45]. The RLR family consists of three homologous protein members: RIG-I, MDA5 and LGP2. Our previous research indicated that MDA5 and LGP2 might play important roles in anti-HIRRV infection defense, which were significantly up-regulated post HIRRV infection both at 10 °C and 20 °C [15]. The expressions of MDA5 in grass carp (Ctenopharyngodon idella) and Japanese flounder were also found to be significantly increased after infections with grass carp reovirus and viral hemorrhagic septicemia virus, respectively [46,47], suggesting that MDA5 plays a key role in the antiviral defense in teleost. LGP2 has also been shown to have antiviral capabilities. The overexpression of Japanese flounder LGP2 significantly delayed the cytopathic effects of VHSV or HIRRV, which is consistent with the results of our study [48]. In this study, significant expression changes of proteins involved in NLR and the RLR signaling pathways were found both at 10 °C and 20 °C after virus infection. When screening for significantly up-regulated proteins and phosphorylated proteins that were specifically highly expressed at 20 °C, we found that some proteins were also annotated to the NLR and RLR signaling pathways. Therefore, the activation of these PRR signaling pathways can effectively respond to HIRRV invasion by inducing the expression of viral response genes, thereby limiting viral replication and regulating adaptive immunity, and such response is more aggressive under 20 °C.

The IFNs induce interferon signaling by binding to the IFN-I receptor to make a broad and potent antiviral response against most viruses that infect vertebrate animals, which leads to the induction of hundreds of ISGs [49,50]. In this study, several ISGs associated with anti-HIRRV infection were identified: interferon-induced protein 44 (IFI44), interferon-induced protein with tetratricopeptide repeats 1 (IFIT1), ISG15 and interferon-induced GTP-binding protein MX1 (MX1). IFI44 plays an important role in respiratory syncytial virus (RSV) infection and is able to reduce RSV genome replication or transcription [51]; however, its role in the antiviral immune process in teleost fish has not been reported. IFIT1 regulates both viral and cellular functions, such as translation initiation, virus replication, double-stranded RNA signaling, cell migration and proliferation, in response to IFN, dsRNA and many viruses [52,53]. It was reported that the IFIT1 mRNA of Japanese flounder was up-regulated after VHSV challenge [54]. Tongue sole (Cynoglossus semilaevis) IFIT1 could be significantly induced by viral infection, which was proved to play a crucial role in antiviral defense by overexpression and knockdown assays [55]. Mx proteins have potent antiviral activity against various RNA viruses [56,57,58]. It has been revealed that the hirame natural embryo cell line (HINAE) stably expressing Japanese flounder Mx protein showed reduced levels of the viral load of HIRRV and VHSV post infection [59]. Poly(I:C) could also effectively induce the expression of ISG15 in Atlantic cod [60]. The quick response of these ISGs upon HIRRV infections indicates important roles for these proteins in antiviral immunity. We found, by qRT-PCR, that the expression levels of these antiviral-related genes were significantly higher at 20 °C than at 10 °C during the early stage of infection. We can speculate that the HIRRV is able to proliferate under different temperature after injection, but the high temperature conditions enabled the fish to activate the antiviral response in a timely and efficient manner, leading to a significant increase in the level of antiviral immunity, which effectively inhibited the virus infection [61]. To be noted, after 48 h, the virus in the low-temperature group continued to proliferate and the high viral load effectively induced an antiviral response in the flounder, while the viral level in the high-temperature group had been suppressed to a very low level and the level of antiviral immune response in the flounder gradually decreased. These results further support that the temperature dependence of HIRRV outbreak was mainly associated with the influence of temperature on the antiviral response of flounder.

The results of KEGG pathway analysis showed that the DEPs and DEPPs were significantly involved in phagosome and endocytosis, and displayed greater activation at high temperatures after viral infection. Endocytosis is the process by which extracellular substances enter the cell through membrane invagination and internalization, where the endocytosis of solid substances is called phagocytosis. Viruses must deliver their genome into the host cells to initiate replication. The majority of viruses enter cells through endocytosis to inject their genetic material into the cytoplasm [62]. When the virus ligands interacted with cellular receptors, it can bind to the cell surface and results in membrane invagination and phagosome formation [63]. In addition, endocytosis plays an important role in the removal of cellular foreign material, thereby protecting the host from viral attack [64]. Innate immune cells specifically recognize and degrade microorganisms by generating phagosomes and phagolysosomes [65]. Phagocytosis relies upon the cell membrane being sufficiently mobile to allow realignment and subsequent engulfment and, so, its function is influenced by the temperature [66]. It was reported that the phagocytosis in channel catfish (Ictalurus punctatus) was significantly inhibited at 10 °C compared with that under 18 °C [67]. However, the specific role played by the phagocytosis in the resistance to HIRRV invasion is not known in flounder. We can speculate that phagocytosis plays a major role in protecting the fish from the virus during HIRRV invasion, resulting in a higher antiviral activity of flounder at 20 °C.

In addition, DEPs under 20 °C are more involved in the focal adhesion relative to 10 °C. Focal adhesion kinase (FAK) is associated with the process of cell adhesion and spreading, which plays a key role in cell signaling pathways [68]. In mammals, several viruses have been identified to regulate FAK activity for its entry or induction of phagocytosis [69]. FAK was identified to interact with IAV nucleoprotein to promote IAV replication by regulating its polymerase activity [70]. In vertebrates, binding of foreign pathogens to integrin-β activates members of the SRC family, thereby recruiting FAK, then regulates the expression of genes associated with phagocytosis and promotes cellular elimination of foreign pathogens [71,72]. Recent data suggest that FAK is a regulator of the mitochondrial anti-viral signal protein (MAVS), which is involved in RIG-I-like receptor-mediated antiviral signaling [73]. In the present study, phagocytosis-related pathways were also clearly annotated, and we suggest that FAK plays a role in activating phagocytic pathways during HIRRV infestation.

It is noteworthy that the most enriched pathway of DEPPs is necroptosis post HIRRV infection under 10 °C compared to the control group. Viruses are specialized intracellular pathogens that require live host cells to multiply. Thus, the host limits replication in virus-infected cells and promotes an antiviral state in neighboring uninfected cells by activating necroptosis pathways [74]. In cells infected by the Influenza A viruses, the necroptotic cell directly eliminated infected cells and mobilized both innate and adaptive immune responses [75]. This result, to some extent, indicates the antiviral status of flounder during the initial stage of HIRRV infection under 10 °C. mTOR signaling pathway is also significantly enriched, and cellular responses to viral infection are influenced by the rapamycin complex 1 (mTORC1), a mechanistic target of the mTOR that drives viral proliferation and survival by regulating anabolic and catabolic processes [76]. Therefore, we hypothesize that HIRRV tries to use that way to prepare materials for its own replication in the host. The mTOR signaling pathway was significantly enriched in the 10 °C virus infection group, which may be the reason for the high mortality rate of flounder at 10 °C. The mTOR signaling pathway was also significantly enriched in the 20 °C HIRRV infection group, while we noticed that the majority of DEPPs were enriched in apoptosis in the antiviral immune response. Viral infection can induce exogenous apoptotic pathways through PRR-dependent expression of programmed death ligand (including TNF-α) [77]. In addition to regulating extrinsic apoptotic signaling, recent evidence suggests that PRRs activate apoptosis through an intrinsic transcriptional non-dependent mechanism [78]. RIG-I and IRF3 are required for the apoptotic process in murine embryonic fibroblasts (MEF) after RNA virus infection or poly(I:C) stimulation [79,80]. Based on these observations, the pathway was subsequently named RIG-I-like receptor-induced IRF3 mediated pathway of apoptosis (RIPA). In the present study, IRF3 was differentially up-phosphorylated at 20 °C, suggesting that it can induce apoptosis in HIRRV-infected cells of flounder under high temperature, thereby inhibiting further virus transmission and thus establishing a broad antiviral response in the early stages of virus infection and resist further invasion of HIRRV.

## 5. Conclusions

The proteome and phosphoproteome analysis revealed that the differential expressed proteins and phosphoproteins were significantly enriched in the immune- and antiviral-related pathways post HIRRV infection. The DEPs and DEPPs involved in the interferon production and antiviral immune signaling pathways including RLR and NLR signaling pathways were significantly enriched post HIRRV infection both under 10 °C and 20 °C, which exhibited stronger and quicker response under 20 °C. This study provided a deeper insight into the differential antiviral responses of flounder to HIRRV infection under different temperatures, which leads a further understanding of interaction between flounder immunity and HIRRV infection.

## Figures and Tables

**Figure 1 biology-12-01145-f001:**
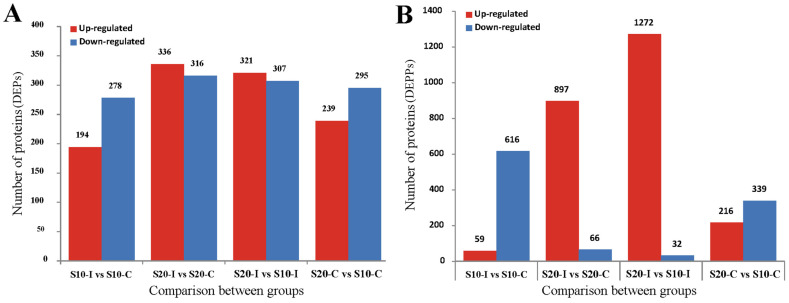
Differentially expressed proteins and phosphoproteins between different two treatments. (**A**): Statistical plot of DEPs; (**B**): quantity distribution of DEPPs and phosphorylated modification sites. Red indicates up-regulation and blue indicates down-regulation.

**Figure 2 biology-12-01145-f002:**
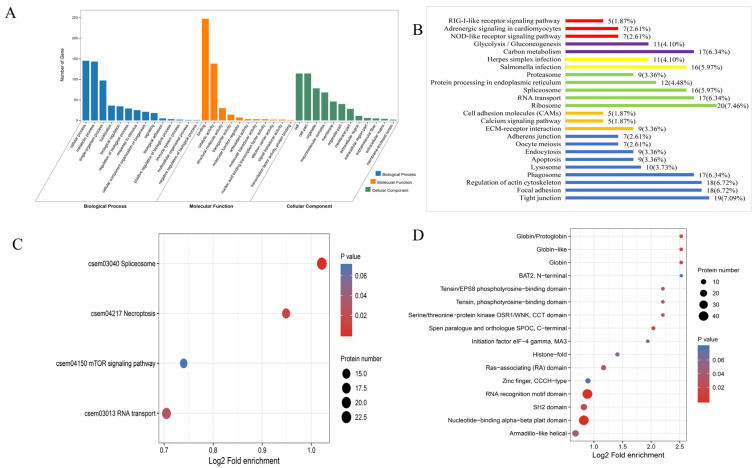
The analysis of identified DEPs and DEPPs under 10 °C post HIRRV infection in flounder. (**A**) Distribution of DEPs in GO functional classification. (**B**) KEGG classification of differentially expressed proteins. The hollow bars represent downregulated proteins, while the solid bars represent upregulated proteins. (**C**) KEGG pathway enrichment analysis of DEPPs. (**D**) Protein domain enrichment analysis of DEPPs.

**Figure 3 biology-12-01145-f003:**
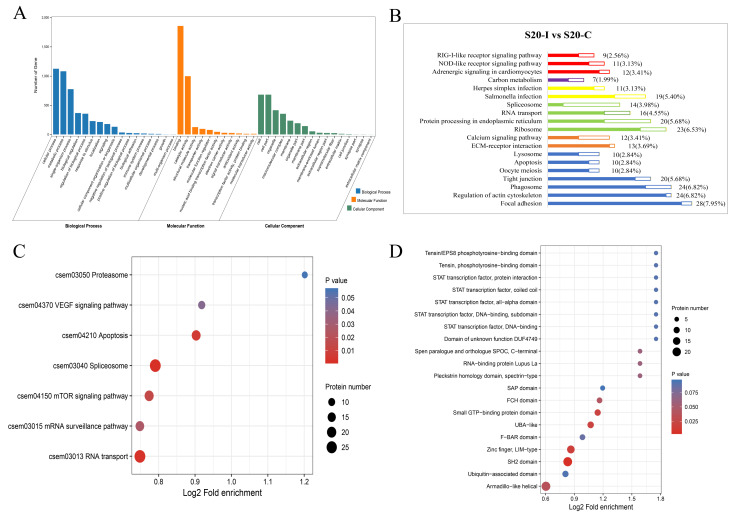
The analysis of identified DEPs and DEPPs under 20 °C post HIRRV infection in flounder. (**A**) Distribution of DEPs in GO functional classification. (**B**) KEGG classification of differentially expressed proteins. The hollow bars represent downregulated proteins, while the solid bars represent upregulated proteins. (**C**) KEGG pathway enrichment analysis of DEPPs. (**D**) Protein domain enrichment analysis of DEPPs.

**Figure 4 biology-12-01145-f004:**
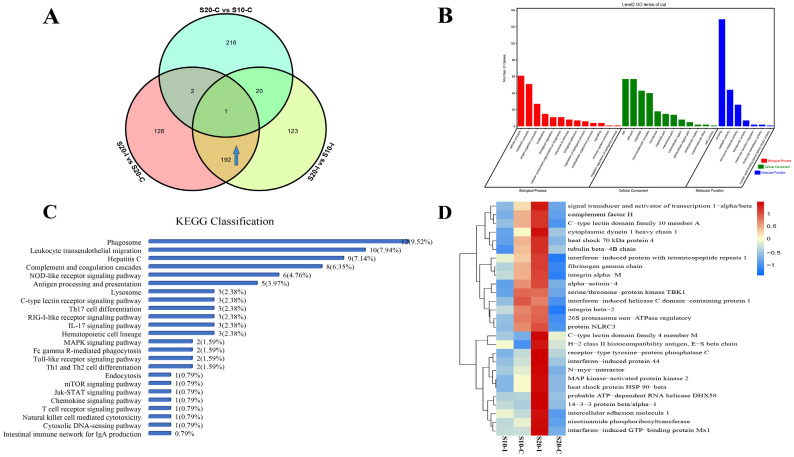
The anti-HIRRV up-regulated proteins affected by temperature. (**A**) Venn diagram of up-regulated proteins between different groups, as shown in the arrow section. (**B**) KEGG classification about the up-regulated proteins involved in immunity. (**C**) GO annotation about the up-regulated proteins involved in immunity. (**D**) Expression patterns of selected proteins under different treatments.

**Figure 5 biology-12-01145-f005:**
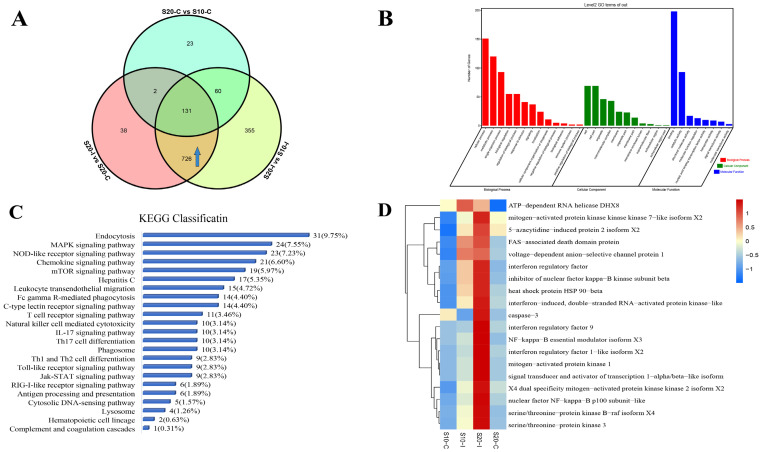
The anti-HIRRV up-regulated phosphorylated proteins affected by temperature. (**A**) Venn diagram of up-regulated proteins between different groups, as shown in the arrow section. (**B**) KEGG classification about the up-regulated phosphorylated proteins involved in immunity. (**C**) GO annotation about the up-regulated phosphorylated proteins involved in immunity. (**D**) Expression patterns of selected phosphorylated proteins under different treatments.

**Figure 6 biology-12-01145-f006:**
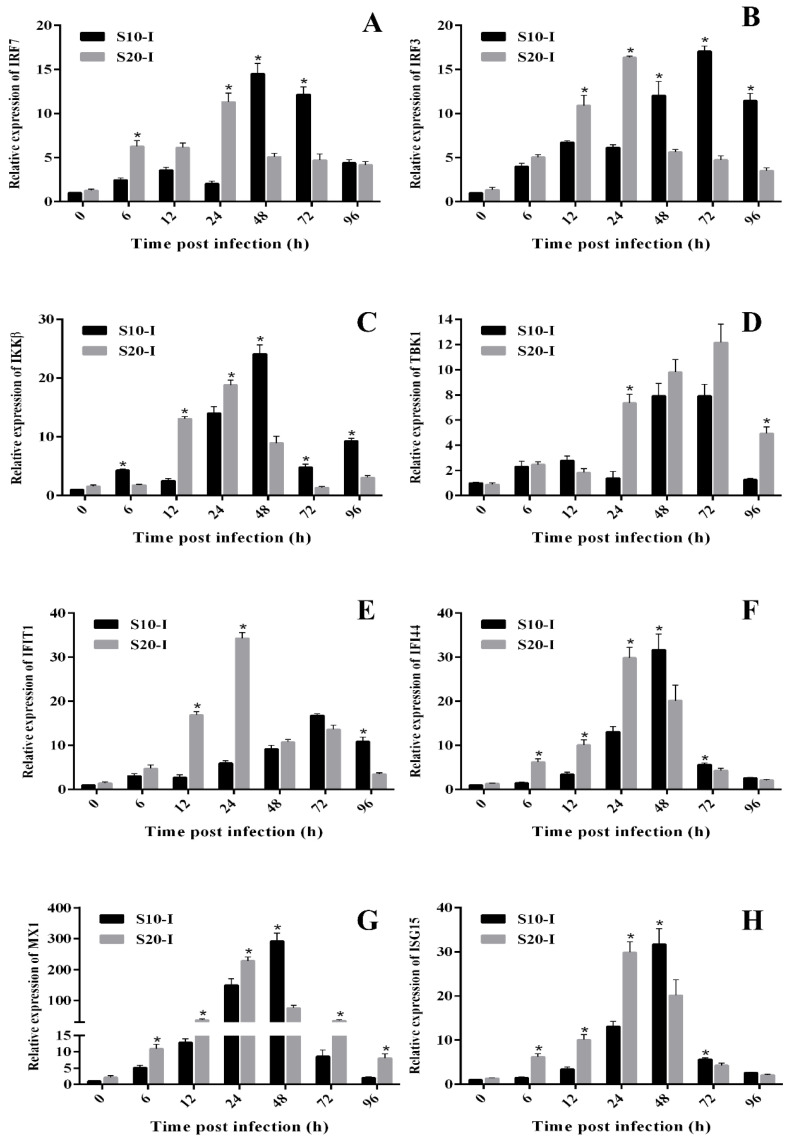
Temporal expressions of genes involved in interferon signaling. (**A**–**H**) Relative expression of IRF7, IRF3, IKKβ, TBK1, IFIT1, IFI44, MX1 and ISG15, respectively. The results were presented as the means ± SEM of five individuals, * *p* < 0.05.

**Table 1 biology-12-01145-t001:** Primers used in this study.

Primers	Sequences(5′-3′)	Length(bp)	Accession No.
TBK1-F	GCAGAGCACCACTAACTACCT	215	XM_020110891
TBK1-R	CAGCGAAGAGCTTCACAAT		
MX1-F	ACCTGCCTGGAATCACC	244	XM_020086975
MX1-R	GACTCCTCTGCTCCTTTG		
IFI44-F	AGTGTTTAACGGACGAG	144	XM_020098597
IFI44-R	CCCAGACCCATAGCA		
IFIT1-F	AGCCACCGAACAGGA	204	KY399812
IFIT1-R	GCCAAAGCAAGAGCC		
IRF3-F	AACTCAAGCCAAACTGACCCG	220	GU017417
IRF3-R	GAAGTCCATAATTCCTCGCACAA		
IRF7-F	ATGGGCAGTGGCAAGTGGT	184	GU017419
IRF7-R	GTTTTCTTGTCTGTGCTCGGTGT		
IKKβ-F	GTCTTGGAGCCTAACG	139	XM_020099249
IKKβ-R	GGGTGCGAGGAGTAA		
ISG15-F	CTCCCATCCAGGTCTTCC	199	AB519717
ISG15-R	GTGCTCTGTGCCTCAACG’		
18S RNA-F	GGTCTGTGATGCCCTTAGATGTC	107	EF126037
18S RNA-F	AGTGGGGTTCAGCGGGTTAC		

## Data Availability

The data in this research are accessible upon demand from the corresponding author.

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
