# Peer review of "Proteomic and Phosphoproteomic Analysis Reveals Differential Immune Response to Hirame Novirhabdovirus (HIRRV) Infection in the Flounder (Paralichthys olivaceus) under Different Temperature"

_biology, 2023, doi:10.3390/biology12081145_

Round 1

Reviewer 1 Report

The manuscript entitled “Proteomic and phosphoproteomic analysis reveals differential immune response to Hirame novirhabdovirus (HIRRV) infection in the flounder (Paralichthys olivaceus) under different temperature” uses a bunch of advanced techniques and high-throughput omics approach to investigate physiological changes in spleen of flounder at two different temperatures, 10℃ and 20℃. The issue considered in the manuscript is interesting and could bring a lot of new knowledge to the field. However, the manuscript is poorly written, clarity and organization are low. But the most important is that the analysis of the results is not appropriate. Therefore, I cannot recommend this manuscript for publication in its current form.

General comments:

1. Materials & Methods section is poorly described. No information was provided about phosphoproteome analysis, and protein extraction was inappropriate for phosphoproteome study. Therefore, this part should be redone or completely removed from the manuscript. There was no information provided about different times of exposure to the virus. However, the results of RT-qPCR are presented for different time points. No information was provided about the number of fish involved in the study and conditions of their well-being during the experiment.

2. The analysis of the results was barely done. Basically, the only conclusion that authors could get from their high-throughput, advanced study is that immune response was increased due to infection, and interferon signaling was stronger at 20℃ than at 10℃. What is the novelty of this study then? The fact that immune response is increased in infected organisms is not new information. I am sure that from such an advanced method as a proteomic study much more useful information could be extracted with potential to be used in remediation efforts to cure or prevent infection. Therefore, the results should be reanalyzed (see specific comments), discussion and conclusion parts should be rewritten completely.

3. In general, my recommendation is to reject the manuscript in its current form, rewrite it, redo or remove phosphoproteome analysis, and resubmit it.

Specific comments:

Line 6: What is this “and” in the end of co-authors?

Line 20: “qRT-PCR assay showed that eight antiviral-related molecules displayed…” What molecules?

Line 55: “…has been reported to infect many many marine and freshwater fish species…” That “many many” is not scientific language.

Line 80: “delayed response of ISGs” explain the abbreviation when mention it for the first time.

Line 83: “…but the viral copies in 10°C infection group were significantly higher…” Number of viral copies, probably?

Lines 123 – 134: What was the number of fish used in the experiment? What were the conditions of exposure (size of tanks, number of fish per tank, water pH, water filtration)?

Lines 137-138: What was the number of fish in each group?

Explain why spleen was the only organ analyzed in this study.

Line 142: “are abbreviated as S10-I and S20-I, respectively, and S10-C and S20-C…” explain abbreviation.

Line 149: For phosphoproteome assay phosphotase inhibitors had to be used for protein extraction. Without those inhibitors phosphoproteome analysis cannot be considered valid. For the protease inhibitor cocktail a producer should be mentioned.

Lines 169-182: No information was provided about identification of post-translational modifications in the proteome.

Lines 228-246: This part provides no useful information for the study and should be removed, including tables 1 and 2. The results should be presented separately for the two temperature groups analyzed. I assume (because it's not obvious from materials & methods) that the protein expression level in infected groups was normalized to the control (not infected) at the same temperature. This information should be added to materials & methods. And differentially expressed proteins should be presented separately for the two groups, 10C and 20C.

Lines 248-253: Provide total description of Figure 1, including comparison of other groups. What about figure 1C? It is not mentioned anywhere in the results. Thus, it should be removed.

Lines 263-264: Were those pathways up- or down-regulated?

Lines 265-267: Were those pathways up- or down-regulated?

Lines 294-295: Were those pathways up- or down-regulated?

Lines 296-298: Were those pathways up- or down-regulated?

Line 314: “…response to the HIRRV infection between under 10℃ and 20℃.” Remove “under”.

Lines 318-320: Up- or down-regulated? Analysis of the results is poorly presented. It seems that there is a major difference between infected groups in terms of up- and down-regulation of immune-related proteins at different temperatures. I would say that this is the main result of this study which is barely mentioned in the description of results. Enrichment of certain pathways is an interesting observation, but without any information on whether the pathway is suppressed or activated, it doesn't provide any useful information for the possible future remediation efforts.

The differentially expressed proteins (between infected group and the control at the same temperature) could be categorized into pathways and pathways should be compared between two different temperatures. It should be clear which pathways are activated and which are suppressed. That could help you to drive a conclusion about the reason why HIRRV infection is more dangerous at 10℃ than at 20℃. You could also compare those pathways or proteins (especially the ones related to immune system) between the control fish at 10℃ and 20℃. Probably, immune system of flounders at 10℃ is just generally suppressed, therefore they have higher risk of being infected.

Figures 1 – 5: Quality of figures is very low, so any review on the content is not possible.

Lines 352-362: There was no information about time-dependency in materials & methods or in the proteome analysis. Moreover, it is stated that fish were sacrificed at 24 hpi. So, how those different groups appeared? In which tissues the gene expression was tested. I suppose this part is from another manuscript and occurred here by mistake.

Figure 6: What genes? What is presented on each panel?

I will not comment on the Discussion and Conclusion as I think these parts should be rewritten completely according to the new analysis of results.

The English language is generally fine, but the overall quality of the manuscript could be improved by professional language editing service.

Author Response

Dear reviewer, 

Thank you for giving us constructive comments on MS titled “Proteomic and phosphoproteomic analysis reveals differential immune response to Hirame novirhabdovirus (HIRRV) infection in the flounder (Paralichthys olivaceus) under different temperature”. These comments were all valuable and very helpful for revising and improving our paper, as well as the important guiding significance to our research. We made a careful revision in which we fully addressed the issues point by point raised from the reviewers’ comments. All modifications were highlighted in yellow in revised MS. And we hope that the revised manuscript is acceptable for publication. The followings are the responses to the comments.

Comment 1

  1. Materials & Methods section is poorly described. No information was provided about phosphoproteome analysis, and protein extraction was inappropriate for phosphoproteome study. Therefore, this part should be redone or completely removed from the manuscript. There was no information provided about different times of exposure to the virus. However, the results of RT-qPCR are presented for different time points. No information was provided about the number of fish involved in the study and conditions of their well-being during the experiment.

Response: Thank you for your comment. We sent the spleen samples to the company for protein extraction and sequencing, thus we are not very clear about the specific process of protein extraction and analysis. After further communication with the company, we provided additional descriptions of the method for phosphoproteome study in “Materials and methods” part (Line 158-159; Line 192-201). Also, the information about the number of fish and their well-being conditions were also supplemented (Line 143-149).

Comment 2:

  1. The analysis of the results was barely done. Basically, the only conclusion that authors could get from their high-throughput, advanced study is that immune response was increased due to infection, and interferon signaling was stronger at 20℃ than at 10℃. What is the novelty of this study then? The fact that immune response is increased in infected organisms is not new information. I am sure that from such an advanced method as a proteomic study much more useful information could be extracted with potential to be used in remediation efforts to cure or prevent infection. Therefore, the results should be reanalyzed (see specific comments), discussion and conclusion parts should be rewritten completely.

Response: Thank you for your valuable and constructive comments. We have categorized the differentially expressed proteins into the up- and down-expressed groups and re-analyze the signaling pathways they are involved in. In our previous MS, the description in the Results and Discussion sections was also focused on these pathways. (Line 256-258; Line 261-262; Line 289-292; Line 296-298; Line 531-533).

Specific comments:

Line 6: What is this “and” in the end of co-authors?

Response: Sorry for the carelessness, the “and” has been deleted.

Line 20: “qRT-PCR assay showed that eight antiviral-related molecules displayed…” What molecules?

Response: The eight antiviral-related molecules were added in the revised MS (Line 20-21).

Line 55: “…has been reported to infect many many marine and freshwater fish species…” That “many many” is not scientific language.

Response: Thank you for your thoroughness and attention to detail. We have deleted one "many" from the MS.

Line 80: “delayed response of ISGs” explain the abbreviation when mention it for the first time.

Response: The full spelling of ISGs was added in the MS.

Line 83: “…but the viral copies in 10°C infection group were significantly higher…” Number of viral copies, probably?

Response: Thank you for the comment, it has been revised in the MS as you suggested.

Lines 123-134: What was the number of fish used in the experiment? What were the conditions of exposure (size of tanks, number of fish per tank, water pH, water filtration)?

Response: Thank you so much for pointing out this issue for us. We used 70 fish in each group for a total of 280. The fish were raised in automatic water cycler system (500L) supplied with aerated and filtrated water under 16°C. Farming water conditions were monitored by the YSI mul-ti-probe system (YSI 556) to meet the following conditions: dissolved oxygen 6.0 ± 0.5 mg/L, pH = 7.0, total ammonia < 0.2 mg/L and nitrite <0.02 mg/L. The detailed information about the culture condition and sampling was supplemented in the revised MS (Line 126-132; Line 143-144).

Lines 137-138: What was the number of fish in each group?

Response: We used 70 fish in each group for a total of 280.

Explain why spleen was the only organ analyzed in this study.

Response: Our previous studies showed that spleen is an important target organ for HIRRV, and a significantly higher viral load was detected by immunofluorescence assay and semi-quantitative PCR [1-2]. It is widely known that the spleen is an important immune organ in fish, and we also found that innate immune pathways were significantly induced by HIRRV infection according to the transcriptome analysis [3]. Thus, we are led to speculate that the spleen plays a crucial role in the anti-HIRRV immune response in flounder. So, the spleen was chosen as the target organ for further investigation in this study.

  • Zhang, J.; Tang, X.; Sheng, X.; Xing, J.; Zhan, W. Isolation and Identification of a New Strain of Hirame Rhabdovirus (HIRRV) from Japanese Flounder Paralichthys Olivaceus in China. J. 2017, 14, 73.
  • Tang X, Qin Y, Sheng X, et al. Generation, characterization and application of monoclonal antibodies against matrix protein of hirame novirhabdovirus (HIRRV) in flounder. Diseases of Aquatic Organisms, 2018, 128(3): 203-213.

[3]  Wang, H.; Tang, X.; Sheng, X.; Xing, J.; Chi, H.; Zhan, W. Transcriptome Analysis Reveals Temperature-Dependent Early Immune Response in Flounder (Paralichthys Olivaceus) after Hirame Novirhabdovirus (HIRRV) Infection. Fish Shellfish Immunol. 2020, 107, 367–378.

Line 142: “are abbreviated as S10-I and S20-I, respectively, and S10-C and S20-C…” explain abbreviation.

Response: I'm sorry we didn't describe it clearly enough for you to be confused about it. For the sake of simplicity, we use "S" stands for "sample", "I" stands for "infection" and "C" stands for "control", so the viral infection groups under 10℃ an 20℃ are abbreviated as S10-I and S20-I, respectively, and S10-C and S20-C respresent two different control groups. Further explanation was made in the revised MS (Line 149-151).

Line 149: For phosphoproteome assay phosphotase inhibitors had to be used for protein extraction. Without those inhibitors phosphoproteome analysis cannot be considered valid. For the protease inhibitor cocktail a producer should be mentioned.

Response: Thank you for reminding us. Due to incomplete knowledge of the company's sequencing process, we overlooked the key reagents in phosphoproteome assay, which has now been supplemented in the revised MS (Line 158-159; Line 192-201).

Lines 228-246: This part provides no useful information for the study and should be removed, including tables 1 and 2. The results should be presented separately for the two temperature groups analyzed. I assume (because it's not obvious from materials & methods) that the protein expression level in infected groups was normalized to the control (not infected) at the same temperature. This information should be added to materials & methods. And differentially expressed proteins should be presented separately for the two groups, 10C and 20C.

Response: Thanks to your valuable suggestions, we have removed Tables 2 and 3 and added information on the normalization process in the material approach (Line 213-214).

Lines 248-253: Provide total description of Figure 1, including comparison of other groups. What about figure 1C? It is not mentioned anywhere in the results. Thus, it should be removed.

Response: I apologize for the lack of detail in the previous description of Fig. 1, which has been described in a more detailed manner for the DEPs and DEPPs identified in each comparison group, and Fig. 1C have been removed.

Lines 263-267; Lines 294-298: Were those pathways up- or down-regulated?

Response: The question you asked is very helpful in improving the quality of our MS. At different temperatures, some proteins in a signaling pathway are activated while others are inhibited before and after HIRRV infection. We made a classification of up- and down-regulated DEPs in each KEGG signaling pathway according to your suggestion and re-drew the Fig.3B. We found that most of the proteins underwent down-regulated expression in 10°C, while up-regulated expression occurs in 20°C, which may explain the stronger antiviral response in fish at 20°C, leading to high survival rates.

Line 314: “…response to the HIRRV infection between under 10℃ and 20℃.” Remove “under”.

Response: Thank you for correcting our grammatical problems, the "under" has been removed from MS.

Lines 318-320: Up- or down-regulated? Analysis of the results is poorly presented. It seems that there is a major difference between infected groups in terms of up- and down-regulation of immune-related proteins at different temperatures. I would say that this is the main result of this study which is barely mentioned in the description of results. Enrichment of certain pathways is an interesting observation, but without any information on whether the pathway is suppressed or activated, it doesn't provide any useful information for the possible future remediation efforts.

Response: To explore how temperature affects the anti-HIRRV immune response in flounder. The DEPs and DEPPs that we are interested in should meet the following criteria. The proteins should be up-regulated post HIRRV infection. Under viral infection state, the expression of proteins should also be positively correlated with temperature, which were up-regulated at S20-I compared to S10-I. Finally, a portion of differentially expressed proteins that were only affected by temperature change were excluded. Because of the low mortality rate of fish at 20℃ and the fact that at 20℃ we identified significantly more up-regulated DEPs and DEPPs than down-regulated. So, in this section we only investigated the role of up-regulated DEPs and DEPPs in the antiviral process.

The differentially expressed proteins (between infected group and the control at the same temperature) could be categorized into pathways and pathways should be compared between two different temperatures. It should be clear which pathways are activated and which are suppressed. That could help you to drive a conclusion about the reason why HIRRV infection is more dangerous at 10℃ than at 20℃. You could also compare those pathways or proteins (especially the ones related to immune system) between the control fish at 10℃ and 20℃. Probably, immune system of flounders at 10℃ is just generally suppressed, therefore they have higher risk of being infected.

Response: Based on your suggestion we have categorized the differentially expressed proteins in the pathway into upregulated and downregulated proteins. We made a classification of up- and down-regulated DEPs in each KEGG signaling pathway according to your suggestion and re-drew Fig.3B and Fig.4B based on the modified data. We found that most of the proteins were down-regulated in 10°C, while up-regulated expression occurs in 20°C, which may explain the stronger antiviral response in fish at 20°C, leading to high survival rates.

Figures 1 – 5: Quality of figures is very low, so any review on the content is not possible.

Response: We apologize for the low quality of the images. Due to the automatic compression of the software, the resolution of the images has been reduced excessively. We have resolved this issue in the revised MS.

Lines 352-362: There was no information about time-dependency in materials & methods or in the proteome analysis. Moreover, it is stated that fish were sacrificed at 24 hpi. So, how those different groups appeared? In which tissues the gene expression was tested. I suppose this part is from another manuscript and occurred here by mistake.

Response: Thank you for pointing out our shortcomings. The detailed information about the sampling and grouping has been supplemented in the revised MS (Line 151-154).

Figure 6: What genes? What is presented on each panel?

Response: We examined the temporal expression of eight genes at different temperatures, panel A to H present the expression profiles of IRF7、IRF3、IKKβ、TBK1、IFIT1、IFI44、MX1 and ISG15, respectively. This information has been added in the figure caption (Line 360-361).

Reviewer 2 Report

The work you have done is notable above all for the laboratory techniques used. The major revision is on the figures that I list below, as they are important and fundamental for verifying what is reported in the text in the results paragraph.

I request a these changes:

in line 55 I believe that the word Many is repeated twice erroneously.

in figure 1 the writing A and B which we find in the caption is missing

Figure 2, figure 3, figure 4, figure 5 which report many of the results must be improved because in this way nothing is read.

I would like to review the work with improved and clear figures in order to be able to read and distinguish what is reported in the text.

Author Response

Dear reviewer, 

Thank you for giving us constructive comments on MS titled “Proteomic and phosphoproteomic analysis reveals differential immune response to Hirame novirhabdovirus (HIRRV) infection in the flounder (Paralichthys olivaceus) under different temperature”. These comments were all valuable and very helpful for revising and improving our paper, as well as the important guiding significance to our research. We made a careful revision in which we fully addressed the issues point by point raised from the reviewers’ comments. All modifications were highlighted in yellow in revised MS. And we hope that the revised manuscript is acceptable for publication. The followings are the responses to the comments.

in line 55 I believe that the word Many is repeated twice erroneously.

Response: Thank you for your thoroughness and attention to detail. We have deleted one "many" from the MS.

in figure 1 the writing A and B which we find in the caption is missing

Response: Thank you for correcting this error for us. Regarding Fig. 1C we did not describe more in our study, so it was removed from the MS as the suggestion of Reviewer #1.

Figure 2, figure 3, figure 4, figure 5 which report many of the results must be improved because in this way nothing is read.

Response: We apologize for the low quality of the images. Due to the automatic compression of the software, the resolution of the images has been reduced excessively. We have resolved this issue in the revised MS.

Round 2

Reviewer 1 Report

I appreciate the efforts that authors made to improve their manuscript. In my opinion the manuscript can be accepted for publication after minor correction.

1. When describing the results of qPCR keep in mind that this method does not analyze a random molecules involved in some pathways (cause it makes readers think that you don't know what you measure). Instead replace the term "molecule" with "mRNA" or "gene expression". These terms are more suitable for a scientific paper. Please, correct throughout the article, especially in the Abstract and Conclusions.

2. Conclusions need to be rewritten to be more sound. Currently, lines 531-534: "Under 20°C, DEPs and DEPPs involved in most of the immune-related signaling pathways showed significant up-regulated expression. It was notable that the proteins and phosphoproteins involved in interferon production and signaling showed stronger response to viral infection under 20°C compared with that under 10°C." basically just repeat the same information. Summarize the most important NEW results obtained in your study in this section. Avoid general phrases, this is not a wikipedia article.

Specific comments: 

Line 256: correct "such as the..."

Line 297: "involved these enriched signaling pathways" correct to "involved in these signaling pathways"

Line 509: "...way to prepare material materials.." remove one of the materials

Author Response

Thank you for giving us valuable comments on MS titled “Proteomic and phosphoproteomic analysis reveals differential immune response to Hirame novirhabdovirus (HIRRV) infection in the flounder (Paralichthys olivaceus) under different temperature”. These comments were all valuable and very helpful for revising and improving our paper. We made a careful revision in which we fully addressed the issues point by point raised from the reviewers’ comments. All modifications were highlighted in green in revised MS. And we hope that the revised manuscript is acceptable for publication. The followings are the responses to the comments.

  1. When describing the results of qPCR keep in mind that this method does not analyze a random molecules involved in some pathways (cause it makes readers think that you don't know what you measure). Instead replace the term "molecule" with "mRNA" or "gene expression". These terms are more suitable for a scientific paper. Please, correct throughout the article, especially in the Abstract and Conclusions.

Response:Thanks for your kind suggestion. All the “molecule” used in qPCR assay were replaced by “mRNA”.

  1. Conclusions need to be rewritten to be more sound. Currently, lines 531-534: "Under 20°C, DEPs and DEPPs involved in most of the immune-related signaling pathways showed significant up-regulated expression. It was notable that the proteins and phosphoproteins involved in interferon production and signaling showed stronger response to viral infection under 20°C compared with that under 10°C." basically just repeat the same information. Summarize the most important NEW results obtained in your study in this section. Avoid general phrases, this is not a wikipedia article.

Response:Thanks for your valuable suggestion. The conclusion was rewritten to make it more specific and sound (Line 521-524).

Specific comments: 

Line 256: correct "such as the..."

Response: Thank you for your meticulous review.  “such as he” has been revised into "such as the...".

Line 297: "involved these enriched signaling pathways" correct to "involved in these signaling pathways"

Response:According to your comment. The "involved these enriched signaling pathways" was corrected to "involved in these signaling pathways".

Line 509: "...way to prepare material materials." remove one of the materials

Response:Thank you for your meticulous review. We have removed one “materials”.

Reviewer 2 Report

-        In the Materials and Methods in paragraph 2.6. you should indicate what the acronyms FDR, DEPs, DEPPs, EPC stand for. I can't find them anywhere.

-        In Materials and Methods paragraph 2.6. you can indicate the criteria with which the InterProScan software assigns / classifies the proteins most present in terms of number to the three functional groups.

-        In the Materials and Methods you can better detail the analysis you carry out with the software for the GO classification and for the KEGG pathway.

-        In figure 1 for graph A in the denomination "Number of proteins" insert (DEPs); in figure 1 for graph B add (DEPPs).

-        I propose to dedicate individual pages respectively to figure 2, figure 3, figure 4, figure 5 as you did for figure 6.

-        Note that again for figure 4 and figure 5 the graphs called B cannot be read even if they are written from the computer. For figure 4, even the writings of figure 4 cannot be read well. The figures must be redone larger, on single dedicated pages, in this way even from the hard copy it is possible to read the results well.

-        The discussion is very long, I propose to shorten it.

-        In the discussion the content from line 393 to line 396 is repeated afterwards from line 422 to line 426.

-        From line 453 to line 455 what has already been said in the results is repeated.

Author Response

Thank you and reviewers for giving us valuable comments on MS titled “Proteomic and phosphoproteomic analysis reveals differential immune response to Hirame novirhabdovirus (HIRRV) infection in the flounder (Paralichthys olivaceus) under different temperature”. These comments were all valuable and very helpful for revising and improving our paper. We made a careful revision in which we fully addressed the issues point by point raised from the reviewers’ comments. All modifications were highlighted in green in revised MS. And we hope that the revised manuscript is acceptable for publication. The followings are the responses to the comments.

  1. In the Materials and Methods in paragraph 2.6. you should indicate what the acronyms FDR, DEPs, DEPPs, EPC stand for. I can't find them anywhere.

Response:Thanks for your reminder. We have provided the full names of the abbreviations FDR, DEPs, DEPPs, and EPC (Line208-212).

  1. In Materials and Methods paragraph 2.6. you can indicate the criteria with which the InterProScan software assigns/classifies the proteins most present in terms of number to the three functional groups.

Response:Thanks for your kind suggestion. We have provided a further explanation on the classification criteria for proteins (Line224-226).

  1. In the Materials and Methods you can better detail the analysis you carry out with the software for the GO classification and for the KEGG pathway.

Response:Thanks for your comment. A detailed information was provided about the analysis for the GO classification and KEGG pathway (Line216-229).

  1. In figure 1 for graph A in the denomination "Number of proteins" insert (DEPs); in figure 1 for graph B add (DEPPs).

Response:Thanks for your good suggestion. The “(DEPs)” and “(DEPPs)” were inserted into denominations of Figure 1.

  1. I propose to dedicate individual pages respectively to figure 2, figure 3, figure 4, figure 5 as you did for figure 6.Note that again for figure 4 and figure 5 the graphs called B cannot be read even if they are written from the computer. For figure 4, even the writings of figure 4 cannot be read well. The figures must be redone larger, on single dedicated pages, in this way even from the hard copy it is possible to read the results well.

Response:According to your suggestion. After adjusting the resolution, Figures 2-5 were formatted individually on separate pages.

  1. The discussion is very long, I propose to shorten it. In the discussion the content from line 393 to line 396 is repeated afterwards from line 422 to line 426. From line 453 to line 455 what has already been said in the results is repeated.

Response:According to your comment. We have further tried our best to shorten the discussion section due to the amount of information obtained from the proteome and phosphoproteome and eliminated redundant content that duplicated the Results section.

Round 3

Reviewer 2 Report

For me the work can be published having made the revisions that I have requested.